# Production and transport mechanisms of NO in the polar upper mesosphere and lower thermosphere in observations and models

Koen Hendrickx[1], Linda Megner[1], Daniel R. Marsh[2], and Christine Smith-Johnsen[3,4]

[1]Department of Meteorology, Stockholm University, Stockholm, Sweden
[2]National Centre for Atmospheric Research, Boulder, Colorado, USA
[3]Section for Meteorology and Oceanography (MetOs), University of Oslo, Oslo, Norway
[4]Norway Birkeland Centre for Space Science, University of Bergen, Bergen, Norway

*Correspondence to:* K. Hendrickx (koen.hendrickx@misu.su.se)

**Abstract.** A reservoir of Nitric Oxide (NO) in the lower thermosphere efficiently cools the atmosphere after periods of enhanced geomagnetic activity. Transport from this reservoir to the stratosphere within the winter polar vortex allows NO to deplete ozone levels and thereby affect the middle atmospheric heat budget. As more climate models resolve the mesosphere and lower thermosphere (MLT) region, the need for an improved representation of NO related processes increases. This work presents a detailed comparison of NO in the Antarctic MLT region between observations made by the Solar Occultation for Ice Experiment (SOFIE) instrument onboard the Aeronomy of Ice in the Mesosphere (AIM) satellite and simulations performed by the Whole Atmosphere Community Climate Model with Specified Dynamics (SD-WACCM). We investigate 7 years of SOFIE observations, covering the period 2007 - 2015, and focus on the Southern hemisphere, rather than on dynamical variability in the Northern hemisphere or a specific geomagnetic perturbed event. The morphology of the simulated NO is in agreement with observations though the long term mean is too high and the short term variability is too low in the thermosphere. Number densities are more similar during winter, though the altitude of peak NO density, which reaches between 102 - 106 km in WACCM and between 98-104 km in SOFIE, is most separated during winter. Using multiple linear regression and superposed epoch analysis methods, we investigate how well the NO production and transport are represented in the model. The impact of geomagnetic activity is shown to drive NO variations in the lower thermosphere similarly across both datasets. The dynamical transport from the lower thermosphere into the mesosphere during polar winter is found to agree very well, with a descent rate of about 2.2 km/day in the 80 - 110 km region in both datasets. The downward transported NO fluxes are however too low in WACCM, which is likely due to missing medium energy electrons and D-region ion chemistry that are not represented in the model.

*Copyright statement.* TEXT

# 1   Introduction

Nitric Oxide (NO) is one of the major background constituents in the lower thermosphere and its presence can have direct and indirect consequences to Earth's radiation budget. NO acts as a natural thermostat in the lower thermosphere (Mlynczak et al., 2003) and the cooling at $5.3\mu$m infrared emission of excited NO is primarily dependent on variations in NO number densities and kinetic temperature (Mlynczak et al., 2005). During polar winter, $NO_x$ species ($NO + NO_2$) can prevail for several days or weeks due to the absence of sunlight and can be dynamically transported to mesospheric and stratospheric altitudes due to the downward motion of the summer-to-winter general circulation (Solomon et al., 1982; Randall et al., 2007). Once in the stratosphere, $NO_x$ catalytically destroys ozone, thereby altering the radiation budget and atmospheric dynamics, and possibly having an effect on surface temperatures. Observational and modelling evidence can be found in this non-comprehensive list: Natarajan et al. (2004); Schmidt et al. (2006); Marsh et al. (2007); Lu et al. (2008); Reddmann et al. (2010); Baumgaertner et al. (2011); Semeniuk et al. (2011); Seppälä et al. (2013); Damiani et al. (2016).

An NO reservoir is present between 100 and 110 km altitude (Siskind et al., 1998; Sheese et al., 2013) and the main production processes of NO involve the interaction of ground state and excited nitrogen with molecular oxygen, while destruction occurs primarily via ground state nitrogen, ionised molecular oxygen and solar UV radiation (Barth, 1995). Several NO chemistry reactions are temperature dependent (Bailey et al., 2002) and NO densities vary with solar and geomagnetic activity. Solar radiation (soft X-rays and UV) is responsible for dissociating the strong $N_2$ and $O_2$ bands, as well are subsequent photoelectrons, while at polar latitudes energetic particle precipitation (EPP) during geomagnetic activity causes this dissociation (Barth et al., 2003). EPP directly affects NO concentrations in the upper mesosphere and lower thermosphere, while it can also indirectly affect stratospheric NO densities via descent of aurorally produced NO (Randall et al., 2007; Funke et al., 2014). Distinguishing between the direct and indirect effects on NO production is difficult and the relative contribution of each is still not determined.

NO is transported from reservoir altitudes into the mesosphere and stratosphere with the downward residual circulation during polar winter, and a strong 27 day periodicity in NO production and subsequent descent into the mesosphere has been observed in SOFIE observations (Hendrickx et al., 2015). A similar response of NO production to recurring geomagnetic forcing every 27 days is seen in SCIAMACHY observations in the upper mesosphere (Sinnhuber et al., 2016). The downward transport is especially prominent in connection to Sudden Stratospheric Warmings (SSW) in the NH winter, after which stratospheric $NO_x$ is strongly enhanced (Randall et al., 2006, 2009; Pérot et al., 2014; Funke et al., 2014, 2017). $NO_x$ can further also be locally produced in the stratosphere by solar proton events (Jackman et al., 2000, 2001; Funke et al., 2011), but these occur infrequently and their direct effect on stratospheric ozone has been found to be half that of the indirect effect (Päivärinta et al., 2016; Sinnhuber et al., 2018). Ensuring a correct representation of EPP effects and a dynamical pathway of NO is essential, since otherwise the flux of $NO_x$ descending in the stratosphere is underrepresented when compared to observations (Shepherd et al., 2014).

Randall et al. (2015) investigated the ability of SD-WACCM to reproduce stratospheric $NO_x$ levels, as compared to observations from HALOE, during a strong SSW and elevated stratopause event in the boreal winter 2003-2004. The $NO_x$ enhancements produced by precipitating auroral electrons were of similar magnitude as in the observations, while the descending flux of this EPP-produced $NO_x$, though present in WACCM, was underestimated by a factor of four. From temperature measurements it was found that WACCM did not properly simulate the SSW recovery and that descent from the MLT into the stratosphere was underestimated. Based on this finding, together with the fact that the simulations only included auroral electrons, the authors concluded that the too low $NO_x$ descent is a combination of missing MEE and insufficient transport from the MLT. The Randall et al. (2015) study shows the difficulty in disentangling the direct and indirect EPP effect on NO, especially during disturbed NH winters.

The EPP indirect effect during the geomagnetically quiet NH winter 2008-2009 has been studied by Funke et al. (2017) to investigate how atmospheric models handle the dynamically active conditions and the associated NO transport. Before the sudden stratospheric warming and elevated stratopause event that winter, $NO_x$ descent was reproduced within 20% of observations, while after the SSW discrepancies became apparent. High-top models, with upper lid above 120 km and including WACCM4, were shown to typically underestimate upper mesospheric temperatures after the elevated stratopause (ES) onset, which manifests itself in a too slow downward transport and too low descending $NO_x$ concentrations. Discrepancies of medium-top models (upper lid around 80 km) with observations are on average smaller but show a large spread, which can be traced back to either the implementation of the gravity wave drag scheme or the prescribed $NO_x$ at the uppermost model layers as constrained from observations. Overall, the authors concluded that atmospheric models were able to represent the EPP indirect effect during the geomagnetic quiet and dynamically active NH winter conditions of 2008-2009, but that improvements could be made with a better dynamical representation of ES events. They further note that during periods of high geomagnetic activity the EPP representation may not be as accurate and that inclusion of MEE could be important.

Similar results were found by Sinnhuber et al. (2018), in which the ability of three global chemistry-climate models to produce stratospheric $NO_y$ in response to energetic particle precipitation was investigated and compared to MIPAS observations during the period 2002-2010. Even though the particle effect is implemented differently in the studied models, the resulting $NO_y$ in the upper mesosphere agrees well between the three simulations. The indirect particle effect, however, is captured rather differently in each model and the resulting $NO_y$ flux that descends into the lower mesosphere and upper stratosphere is dependent on the timing of the downwelling and rate of descent.

The occurrence of polar vortex breakups during SSW events and accompanied reformation of the stratopause region in the northern hemispheric winter complicates the polar vortex descent (Randall et al., 2015; Funke et al., 2017; Orsolini et al., 2017) and the contribution of MEE during geomagnetic active conditions imposes further difficulties by impacting both the direct and indirect EPP effect on NO densities. Smith-Johnsen et al. (2017) disentangle the (in)direct EPP effects on Antarctic NO during a 2010 geomagnetic storm by using a continuous energy spectrum for precipitating electrons between 60 and 120 km. They found that during that particular event NO variability above 90 km could be up to 95% accounted for by the direct EPP

effect, while only 35% or less could be attributed to direct EPP below 80 km.

In this work we study the general production and transport of NO. Since SSW events during the NH winter complicate the typical polar vortex descent and create an extra downward draft during the recovery phase, we choose to focus on the Antarctic MLT region, where SSW generally do not occur. We first compare the climatological NO observations from SOFIE and simulations from SD-WACCM in the lower thermosphere and mesosphere (Sect. 3.1). The physical drivers of NO are investigated in Sect. 3.2 for both model and observations using multiple linear regressions. We then investigate the winter transport of NO enhancements after geomagnetic disturbances in Sect. 3.3 and derive a polar vortex descent rate in the MLT region, from which we determine the contribution of MEE to the NO fluxes. The results are discussed in Section 4 and in Section 5 conclusions are given.

## 2  Datasets

### 2.1  AIM/SOFIE

Since May 2007, the SOFIE instrument on board the AIM satellite has performed atmospheric profile scans 15 times a day, to obtain vertical distributions of temperature, ice water content and trace gases (NO, $CO_2$, $CH_4$ and $O_3$) (Gordley et al., 2009). NO volume mixing ratios (VMR) are retrieved using the $5.3\mu$m absorption band, with an approximate vertical resolution of 2 km. The AIM satellite is in a retrograde, sun-synchronous, polar orbit. Since SOFIE uses the solar occultation technique, the local sunrise and sunset measurements in the Southern hemisphere (SH) and Northern hemisphere (NH), respectively, are limited to a latitudinal coverage from $65°$ to $85°$, depending on the time of year. Due to the orbital drift of AIM (from mid 2012 onward) the latitudinal coverage is drifting towards lower latitudes with time. The effective latitudes covered in this study range from $83°$S to $50°$S with a semi-annual periodicity and with the more poleward latitudes taken during the equinoxes and the more equatorward latitudes during solstices.

The NO profiles are reported from 35 km to 150 km on a 200 m altitude grid and are available on the SOFIE website (sofie.gats-inc.com). In this study, daily averaged NO (v1.3) values in both VMR and number density are used and a further vertical smoothing of the NO data with a 2 km low pass filter is applied. An empirical correction to the NO VMR data is applied as described by Gómez-Ramírez et al. (2013). To investigate long and short term variations at high latitudes, all available data from 20 May 2007 to 1 February 2015 are used. During local summer, polar mesospheric clouds (PMC) influence the observation at the $5.3\mu$m band and cause higher NO concentrations at and below PMC height. No correction is available as of this writing and we therefore neglect NO retrievals during PMC season (from day of year (DOY) 315 to DOY 53) in our comparison to WACCM.

## 2.2 SD-WACCM

This study uses the NCAR Community Earth System Model with WACCM (Marsh et al., 2013) as its atmospheric component. The model has 88 pressure levels from the ground to about $5.9 \times 10^{-6}$ hPa. For comparison to observations, we determine for each geopotential height $H$ the geometric altitude $Z$, following

$$5 \quad Z = \frac{r_{\text{Earth}} H}{r_{\text{Earth}} - H}, \tag{1}$$

with $r_{\text{Earth}}$ the Earth radius, and interpolate onto a fixed altitude grid up to 140 km with 2 km vertical resolution. The horizontal resolution is $1.9°$ latitude by $2.5°$ longitude and the timestep is 30 minutes. Output is written as the simulation runs and represents the model value at the nearest latitude, longitude and UT of the SOFIE observation profile. The model provides volume mixing ratios $NO_{\text{VMR}}$, which are converted into number densities using the ideal gas law equation:

$$10 \quad NO_{\text{den}} = \frac{P}{kT} NO_{\text{VMR}}, \tag{2}$$

with $P$ and $T$ the respective simulated pressure and temperature and $k$ the Stefan-Boltzmann constant. The simulations used in this work are performed with specified dynamics (SD-WACCM), relaxing horizontal winds and temperatures to data from the Modern-Era Retrospective Analysis for Research and Applications (Rienecker et al., 2011) in the troposphere and stratosphere, with a free-running atmosphere above 60 km. The simulations follow the reference chemistry climate model initiative (REF-C1SD) forcing scenario from the SPARC Chemistry Climate Model Initiative (Eyring, 2013). Solar fluxes are from the Naval Research Laboratory (NRLSSI v.1) empirical solar model and vary daily, while the parametrised aurora varies with the daily Kp index. The model is run with enhanced eddy diffusion (Prandtl number 2) as this enhances the rate of eddy diffusion (Smith, 2012) and improves trace species concentrations in the MLT region (Garcia et al., 2014). A control simulation with Prandtl number 4 is used as a sensitivity test. The Nitric Oxide Empirical Model (NOEM) is used as an upper boundary condition for modelled NO concentrations (Marsh et al., 2007) and is based on 2.5 years of observations made by the Student Nitric Oxide Explorer (SNOE) satellite during the inclining phase of solar cycle 23 (Marsh et al., 2004).

## 3 Results

This section is divided into three parts, starting with similarities and differences in MLT NO between SOFIE and WACCM. In Section 3.2 the relative importance of the physical drivers of NO is investigated while in Section 3.3 the dynamical aspect of EPP-produced NO is compared.

### 3.1 NO in the mesosphere - lower thermosphere

A seasonal climatology of the Antarctic NO in number density and volume mixing ratio (VMR) is shown in Fig. 1 and Fig. 2 respectively, for both SOFIE and WACCM data. The observing latitude is closer to the polar regions during winter and summer observations, as described in Section 2. In Fig. 1 the total number density of SOFIE observations show the NO reservoir to

be at approximately 100 km, with changes throughout the year in the altitude of the maximum density. Typical polar vortex descent can be seen in the Antarctic winter from March through September. The enhanced NO densities around 85 km during summer are an artefact in the data product due to enhanced radiation in the observed NO band in the presence of noctilucent clouds. It is clear that WACCM simulates the NO reservoir at a higher altitude and with an overall higher column density in the lower thermosphere. Below the mesopause region, a strong seasonal cycle is present and WACCM tends to underestimate the NO number densities, particularly during winter, as compared to SOFIE. Figure 2 shows a similar climatology in NO VMR with a six order of magnitude change in the considered altitude range. The climatological mesopause altitude in each dataset is also shown with a white contour line Fig. 2. It varies between 86-98 km in SOFIE and between 78-100 km in WACCM data, while the SOFIE mesopause is typically 4 km lower during winter and 4 km higher during summer than the WACCM mesopause. During summer and winter the WACCM mesopause is up to 10 K colder than SOFIE, while being warmer during the equinoxes (not shown).

Figure 3 shows in more detail how the altitude of the NO maximum changes throughout the year. For SOFIE data the NO maximum ranges in altitude between $100 - 102$ km in summer and early winter to $96 - 100$ km during mid winter. At the end of winter and in early spring, the mesospheric overturning circulating winds change direction and the altitude of the NO maximum layer increases up to $104$ km before restoring to around $100 - 102$ km. This altitude is lower than the commonly accepted peak altitudes of $105 - 110$ km (see e.g. Solomon et al. (1999); Siskind et al. (1998); Dobbin et al. (2006)) but is in agreement with NO observations from for example the sounding rocket project ECOMA (Hedin et al., 2012), ACE-FTS satellite observations (Sheese et al., 2013) and the OSIRIS and SMR instruments onboard the Odin satellite (Sheese et al., 2011). During Antarctic summer, WACCM simulates the peak density at similar altitude levels as SOFIE. However, during winter the NO maximum is at an altitude of $104$ km, down from $106$ km, where the NO peak densities are found during the equinoxes. NO descend during spring to winter bridges about 4 km in altitude in SOFIE and 2 km in WACCM. It can also be seen from Fig. 1 that the WACCM total density in the thermosphere is higher around the equinoxes in March and September than during summer or winter. Equinoctial geomagnetic activity maxima have long been recognized to occur (Russell and R. L., 1973; Lyatsky et al., 2001) and could be a possible reason for the NO enhancements in WACCM during these periods. Therefore, the discrepancy of equinoctial NO between SOFIE and WACCM in the lower thermosphere could be an indication that the model is too sensitive to changes in geomagnetic activity.

A key aspect of understanding differences between model and observations is how much NO is present in the lower thermosphere throughout the year. Figure 4 therefore shows the mean NO density between 90 and 140 km altitude. WACCM NO densities are on average 1.6 times higher than in SOFIE, whereas in summer it is twice as much. During winter the difference becomes smaller (a factor 1.2). Another approach to investigate the lower thermosphere NO densities is to compare the mean density around the NO maximum. The peak NO density in WACCM is situated between 102 and 106 km while in SOFIE it is between 96 and 104 km altitude. By comparing the NO average over a 10 km region centred around the altitude of peak NO one minimises differences introduced by, for example, atmospheric dynamics. The right hand panel in Fig. 4 shows the

evolution of this climatological 10 km average. One can see that WACCM still has more NO: on average 1.4 times as much as SOFIE, ranging from similar winter values to 1.8 more summer values. It should also be noted that apart from the higher NO column densities, the seasonal variation within each dataset is different: in SOFIE observations winter values are 3.5 times larger than summer values, while the winter-summer ratio is a factor of two in WACCM. Seasonal variability of the NO profiles are highlighted in Fig. 5 and reveal that above 100 km WACCM produces too high NO concentrations in the climatological mean.

Since we are interested in NO densities during the dynamical coupling of the MLT region, we conclude this section by showing winter year to year variability of NO profiles in Fig. 6, which highlights structural differences between the observations and model. The winter is here defined as a 90 day period centred at the June solstice. A large year to year variation is present in the observations with NO values during winter 2013 being three times larger than during winter 2009. This in contrast to the model data in which significantly less variation is found from year to year with a maximum difference of about a factor 1.25. In years with low geomagnetic activity, NO concentrations are considerably overestimated by WACCM while they are underestimated in years with high geomagnetic activity. The inter-annual variability of winter NO concentrations thus follows the level of geomagnetic activity more closely in SOFIE data than in WACCM, with an overall too high background of WACCM NO in the lower thermosphere.

We have so far thus found that WACCM simulates higher NO values at higher altitudes in the lower thermosphere and with less yearly and seasonal variations when compared to SOFIE observations. Plausible reasons for the obtained differences are: a too small NO flux is transported downward during the Antarctic winter, an incorrect meridional gradient of NO revealed by a seasonal shift of the observing latitudes, too much NO production and/or too little NO destruction in the lower thermosphere. The excess summer time NO as compared to SOFIE indicates that the production or destruction mechanisms of NO in WACCM may not be entirely correct. In the next section we will first investigate the drivers of NO variability and how well they agree between model and observation, while in Section 3.3 we will investigate the dynamical picture of winter NO.

## 3.2 Physical drivers of NO

As described in the introduction, solar radiation (soft X-rays and UV irradiance) and photoelectrons ionise and dissociate the main constituents present in the lower thermosphere ($O, O_2, N_2$) creating the elements for NO chemistry to take place. At polar latitudes precipitating energetic particles have a similar effect. The multiple linear regression (MLR) method can been used to determine coefficients for solar and geomagnetic variability, which are related to NO concentrations (Marsh et al., 2004; Bender et al., 2015). The relative importance and contribution of each physical driver to the NO budget in the lower thermosphere can be determined in a similar MLR approach. Hendrickx et al. (2017) obtained consistent results between NO observations from SOFIE and SNOE even though observations were separated nearly a decade in time and the former instrument uses solar occultation while the latter uses UV spectrometry. A similar analysis performed on SOFIE and WACCM data can show whether the correct processes drive NO densities at high latitudes in the model. Since the seasonal NO climatology represents

a mode of variation that we do not seek to explain, we deseasonalise the datasets by subtracting the seasonal climatology and focus on the direct production and destruction mechanisms. Figure 7 reveals that between $70\%$ and $85\%$ of the NO budget can be explained by the climatology shown in Fig. 1 and that throughout the lower thermosphere the WACCM climatology can explain a larger portion of the NO density than the SOFIE climatology. This is a result of the low year to year variability in the model. The remaining variations in the NO anomalies are then driven by variability in geomagnetic activity and solar irradiance upon which they are regressed:

$$\Delta\mathrm{NO}(z, \mathrm{AE}, \mathrm{Ly}\alpha, t) = \gamma_{\mathrm{AE}}(z)\mathrm{AE}(t) + \gamma_{\mathrm{Ly}\alpha}(z)\mathrm{Ly}\alpha(t) + \epsilon(z, t), \tag{3}$$

where $\gamma_{\mathrm{AE}}$ and $\gamma_{\mathrm{Ly}\alpha}$ are the estimated coefficients of the corresponding geomagnetic Auroral Electrojet (AE) index and solar Lyman-$\alpha$ (Ly$\alpha$) irradiance regressors, $\epsilon$ is the residual error term and $\Delta\mathrm{NO}$ denotes the anomaly of NO from its climatological value. More information in Hendrickx et al. (2017).

The MLR output combined with the climatological contribution results into a total explained NO variance larger than $90\%$ for both SOFIE and WACCM (see Fig. 7). The altitudinal profile of the MLR estimated coefficients is shown in Fig. 8. Geomagnetic activity impacts the NO variations in a similar way in both datasets with the highest contribution above 110 km. The parametrised auroral input in WACCM deposits most of the energy above 100 km and the larger difference between the SOFIE and WACCM geomagnetic impact below 105 km is therefore likely due to missing medium energy electrons. The estimated $\gamma_{\mathrm{AE}}$ coefficient in WACCM shows a similar shape as the estimated coefficient in SOFIE, but is slightly smaller in value below 120 km, which can explain the lower year-to-year variability in WACCM NO that was seen in Fig. 6. Throughout the lower thermosphere a small to negligible impact of solar irradiance is to be expected at high latitudes as solar soft X-rays and EUV are most important for NO production at equatorial latitudes. Variations in polar NO attributed to solar irradiance in SOFIE observations are small and consistent with zero below 115 km and become slightly negative above that altitude. The effect of irradiance in WACCM data seems to be more pronounced at high altitudes and differs significantly from the SOFIE irradiance impact, suggesting that solar forcing due to soft X-rays or UV photolysis has a stronger effect on WACCM NO than on what is observed.

To investigate the effect of solar irradiance further, one can rewrite Eq. (3) to

$$
\begin{aligned}
\mathrm{NO}_{\mathrm{model}} &= \mathrm{NO}_{\mathrm{clim}} + \Delta\mathrm{NO} \\
&= \mathrm{NO}_{\mathrm{clim}} + \overline{\Delta\mathrm{NO}} + \gamma_{\mathrm{Ly}\alpha}\frac{\sigma_{\Delta\mathrm{NO}}}{\sigma_{\mathrm{Ly}\alpha}}\left(\mathrm{Ly}\alpha - \overline{\mathrm{Ly}\alpha}\right) + \gamma_{\mathrm{AE}}\frac{\sigma_{\Delta\mathrm{NO}}}{\sigma_{\mathrm{AE}}}\left(\mathrm{AE} - \overline{\mathrm{AE}}\right),
\end{aligned}
\tag{4}
$$

with $\overline{\Delta\mathrm{NO}}$ and $\sigma_{\Delta\mathrm{NO}}$ the mean and standard deviation of NO variations to scale to zero mean and unit variance (similar for AE and Ly$\alpha$) , and with $\mathrm{NO}_{\mathrm{clim}}$ the seasonal climatology. The sign of the estimated coefficient needs to be considered together with the time evolution of the regressor, as the AE and Ly$\alpha$ variations can be both positive and negative. The contribution of radiation to the NO density can thus be identified as the third term in Eq. (4) and is shown in Fig. 9. At lower altitudes where

$\gamma_{\mathrm{Ly}\alpha} > 0$ and when solar activity is below average (solar minimum conditions) the contribution to NO will be negative. Above average solar activity (solar maximum) will contribute to more NO. At higher altitudes $\gamma_{\mathrm{Ly}\alpha}$ is negative and the opposite is true: during solar minimum years the effect of radiation is to enhance NO concentrations, while at solar maximum years a lowering effect is seen. A positive sign of the estimated coefficient does therefore not necessarily mean production at that altitude since the whole term needs to be considered: in a time period when Ly$\alpha$ is below average it either means destruction or less production than normally.

The NO contribution due to solar radiation has clearly a larger effect on WACCM NO than on SOFIE NO at 130 km. The impact, however, seems to be dependent on the phase of the 11 year solar cycle. To test this assumption an MLR is performed with the Ly$\alpha$ regressor replaced by its third-order polynomial fit, without small day-to-day variations. A similar profile of the estimated coefficient $\gamma_{\mathrm{Ly}\alpha}$ was obtained throughout the lower thermosphere. This implies that it is not the shorter term smaller variations in Ly$\alpha$ that are causing the NO variations, but rather the variations on long timescales, similar to the 11 year solar cycle. It could also imply that the high latitude NO densities are not varying with irradiance changes, but rather with a process in the lower thermosphere that follows the 11 year solar cycle, such as for example temperature (Gan et al., 2017). This was also suggested by Marsh et al. (2004) to explain a negative contribution of solar variability at high latitudes.

Figure 9 also shows the NO contribution due to solar radiation at 130 km in NOEM. This NOEM output is on similar magnetic latitudes as SOFIE observations and is offset by a factor $5.10^6$ cm$^{-3}$ because it acts on a different climatological background than the MLR. The solar induced NO in NOEM behaves very similar to that in WACCM, even though the radiation component in the MLR is linear with Ly$\alpha$ and logarithmic with F10.7 in NOEM, and shows the same long term trend. Because NOEM is used as an upper boundary condition for NO at the WACCM model top, discrepancies between WACCM and SOFIE at this altitude are likely caused by differences between NOEM and SOFIE. At 100 km, the solar contribution to NO in WACCM and SOFIE agree very well, which implies that the chemistry in WACCM reacts similarly to UV variability as in the observations. The contribution of radiation in NOEM at 100 km is of opposite sign (not shown) because the associated EOF is negative (Marsh et al., 2004). This implies that, since SOFIE and WACCM show a similar variation, NOEM did not properly capture the radiation impact at these lower altitudes from the shorter SNOE dataset.

## 3.3 Dynamical transport of NO

In winter, dissipating gravity waves cause turbulent mixing and an effective transport of air from the lower thermosphere into the mesosphere, thereby creating a pathway for NO to descent from the thermospheric reservoir down into the middle atmosphere where it can destroy ozone. Perturbed geomagnetic activity periods will create enhanced NO densities, which are transported down into the polar vortex. Following Hendrickx et al. (2015), we perform a superposed epoch analysis (SEA) on SD-WACCM winter data to compare the model and observational response of NO after increased geomagnetic activity.

A SEA was performed on dates on which geomagnetic activity, as represented by the AE index, showed increases that were larger than 2 standard deviations of the dataset. The dates are given in Table 1 and correspond to a doubling of normal geomagnetic activity. The resulting NO responses are enhancements from a running monthly mean and reveal the 27 day periodicity of NO production, shown in Fig. 10. On the central epoch date, SOFIE observes NO increases up to $80\%$ while increases reached in SD-WACCM are much smaller, up to $35\%$. Similarly, SOFIE NO enhancements are larger for the recurring dates 27 days earlier and later.

To study the rate of downward transport we identify at which altitude the maximum NO enhancement is situated. Figure 11 reveals that the NO increase starts at 105 km in SOFIE and 112 km in WACCM and that progressively with time, WACCM almost consistently places the NO enhancements 5 km higher than SOFIE. The descent rate of the NO peak enhancements is thus about 2.2 km/day in both datasets. An epoch analysis on the WACCM control run with standard diffusion (WACCM Pr4) shows that the NO enhancements descend with a rate of about 2.1 km/day. The increases in absolute densities are shown in the right hand side of Fig. 11 and indicate that the maximum enhancements are lagged by two days from the geomagnetic onset, and that SOFIE observes double the increase as compared to WACCM. Maximum values exponentially decrease with time and the difference between SOFIE and WACCM becomes progressively larger lower in the atmosphere. After 13 days the difference reaches a factor 4 with the enhanced diffusion run and a factor 9 with the standard diffusion run. Even though enhanced diffusion decreases the differences between SOFIE and WACCM in descending NO fluxes, a factor 4 difference remains, despite the similar inferred rate of descent. This implies either missing NO production, too much NO destruction or horizontal diffusion in the model. A possible source of NO that is not included in the current model is ionisation by medium energy electrons (MEE) (Arsenovic et al., 2016) and D-region ion chemistry (Andersson et al., 2016).

Another way to study how much NO is being transported downward is to calculate the percentage that remains from a specific altitude level. Because WACCM places the NO enhancements 5 km higher than SOFIE and dynamics are different at different altitudes, we study the percent NO that remains once the enhancements passed the 100 km altitude level. Density enhancements in SOFIE NO pass this level at day 2.4 and on day four $72\%$ of the NO enhancement at 100 km remains as can be seen in Fig. 12. For WACCM the enhancements reach the 100 km level at day 4.5 and on day six only $67\%$ remains. At about 97 km altitude there is therefore an NO deficit of around $5\%$. Extending this process to lower altitudes gives an indication of how this deficit varies throughout the upper mesosphere. The middle panel of Fig. 12 shows the inferred difference between SOFIE and WACCM for every kilometre between 80 km and 100 km, revealing that the deficit ranges between $2\%$ and $9\%$ and maximises around 90 km. This is an indication that a process is missing in the model, which can produce differences up to $9\%$ with the observations in the NO descent. Altering the arbitrary altitude of 100 km up or down does not change the range of deficit percentages nor the level where it maximises.

A production mechanism of NO that is not included in this version of WACCM is MEE. The selected events for the SEA are during strong geomagnetic activity and can therefore be considered to include MEE. A similar SEA is performed on 66

dates where geomagnetic activity was enhanced but not to its most active levels (variations between $1\sigma$ and $2\sigma$), ensuring NO production but minimising MEE. The descent rate of the maximum NO enhancements is similar for SOFIE (2.1 km/day) and WACCM (2.3 km/day) in the 80 - 110 km altitude region (not shown). The time evolution of the NO percentage after it passed the 100 km altitude level is shown in the right panel of Fig. 12. The inferred difference between SOFIE and WACCM for the

medium storms is also shown in the middle panel of Fig. 12 and reveals that the deficit now reaches up to $5\%$. This implies that the EPP indirect effect on NO can have a contribution of $4\%$ of direct NO production by MEE. Because the epoch analysis was performed on dates with moderate geomagnetic activity, the occurrence of MEE was minimised but not excluded: the MEE contribution we determined is therefore an effective lower limit. The remaining difference could be related to non-excluded MEE or D-region ion chemistry.

## 4    Discussion

The simulated Antarctic NO densities in WACCM display the general features of NO in the mesosphere and lower thermosphere as observed by SOFIE. However, there are several differences. WACCM produces higher NO average concentrations throughout the lower thermosphere, with a lower year to year variability and higher altitude of peak NO density.

The results of the MLR indicated that NO variations are determined by geomagnetic activity and solar radiation. The impact of solar radiation however seems to be dependent on the phase of the 11-year solar cycle and it effects WACCM NO more strongly than is observed by SOFIE. Since the variations in NO as observed by SOFIE and SNOE behave in a consistent way (Hendrickx et al., 2017), the result shown in Fig. 8 indicates that the UV/EUV radiation, as represented by the Ly$\alpha$ regressor, appears to have a stronger impact in WACCM NO than in the observations. As argued above, this could be related to tempera-

ture changes. WACCM uses the NO concentrations from NOEM as an upper boundary condition (Marsh et al., 2007). NOEM is a model which is based on 2.5 years of SNOE measurements taken during the ascending phase of solar cycle 23 and is able to reproduce about $50\%$ of the variance of all SNOE observations (Marsh et al., 2004). Climatological NO densities simulated by NOEM and WACCM were compared (not shown) and it was found that both models vary very similarly in concentration, altitude of NO peak, thermospheric NO profile and year to year variation. Because the contribution of solar radiation to the NO

budget at 130 km behaves in a similar way in NOEM and WACCM, it implies that WACCM at its upper altitudes is strongly constrained by NOEM and that differences between WACCM and SOFIE at these altitudes are likely caused by differences between NOEM and SOFIE.

Throughout the lower thermosphere and during all seasons, higher NO concentrations are present in WACCM. NO concen-

trations are very sensitive to the branching ratio of excited and ground state nitrogen $P(N(^2D)/N(^4S))$ during $N_2$ dissociation (Barth, 1995). WACCM has a constant branching ratio of 0.60 which means that 60% of atomic nitrogen is produced in the excited state (Marsh et al., 2007). As $N(^2D)$ is the primary source and $N(^4S)$ the primary loss of NO, one possibility of the higher WACCM NO concentrations is that a too high of a branching ratio results into more NO production and less destruction.

Determining rates and branching ratios in several reactions of the NO chemistry is challenging and large uncertainties remain: some studies, for example, have suggested a ratio of 0.5 (Solomon et al., 1982) while recent research advises an altitude dependent ratio ranging 0.50 at 90 km to 0.60 at 150 km (Yonker, 2013). A second possibility that further could alter the sensitivity of the NO chemistry to solar radiation is the temperature in the lower thermosphere, which impacts temperature dependent reactions. Furthermore, simulating correct atomic oxygen concentrations in the lower thermosphere is also of importance. Yet another possible solution may be related to outdates values of reaction rates or missing reactions, see Yonker (2013) for a recent update. A detailed analysis of which reactions could be updated is outside the scope of this study, but would be valuable future work to make improvements in NO modelling.

The general features of the thermospheric response during the 5 April 2010 geomagnetic storm were rather accurately simulated by the coupled ionosphere-thermospheric TIEGCM model, although differences with observations remained in for example the NO cooling rate (Sheng et al., 2017). The authors found that the differences in NO cooling power between TIEGCM and TIMED/SABER observations were improved by obtaining larger NO number densities, which they accomplished via a new temperature dependent reaction rate for the $N(^2D) + O_2 \rightarrow NO + O$ reaction. An excess of thermospheric NO as compared to satellite observations is present in WACCM, as found in this study. Given that the TIEGCM and WACCM models share a similar implementation of the thermosphere, it is likely that TIEGCM also has an NO excess. In that case an increase in NO densities would appear not to be a solution to improve NO cooling rates.

Another key aspect is the NO descent in the MLT region during polar winter, since the $NO_x$ flux that is transported into the lower mesosphere and stratosphere is important for catalytic ozone destruction and atmospheric dynamics. Differences in atmospheric dynamics and the size or location of the polar vortex between observations and simulations could introduce additional variation in the SOFIE - WACCM comparison. However, a SEA performed on geomagnetic active dates revealed that NO enhancements decrease in altitude with the same descent rate (about 2.2 km/day) in the 80 to 110 km altitude region in WACCM and SOFIE. The MLT descent in the SH therefore does not seem to suffer from dynamical disturbances, as it does in the NH. Eddy diffusion is the driving force of downward transport of trace species and is enhanced in this version of WACCM by halving the Prandtl number to 2. In previous versions, WACCM used a Prandtl number of 4 and halving it was shown to improve the comparison of MLT region CO and $CO_2$ between model and satellite observations (Garcia et al., 2014). A control run with Prandtl number 4 confirms that the descent rate is slightly lower (2.1 km/day) and that the descending NO flux is considerably less (about half) after two weeks.

However, even though the rate of descent of the NO enhancements is the same, the absolute increases in WACCM and SOFIE are different. The MLR shows that the impact of geomagnetic activity on NO variations is similar in both datasets, while the NO enhancements obtained after the SEA show a larger increase in the observations. This is interesting and may seem contradicting at first. The SEA shows the direct impact of geomagnetic activity and reveals the NO response after 17 strong AE events. The MLR on the other hand highlights the impact of drivers on a daily basis and therefore gives a relatively

high weight to the more commonly occurring small variations. The different NO response is therefore most likely related to the intensity of the geomagnetic events and could perhaps be linked to a non-linear response to auroral input (Barth, 1995; Bailey et al., 2002).

In the light of the HEPPA-II intercomparison project, Funke et al. (2017) performed an evaluation of the dynamically active NH winter of 2008-2009 as observed by 7 satellites and simulated by 8 atmospheric models. The authors concluded that the EPP indirect effect was adequately described in the models and that inclusion of MEE in one of the models (HAMMONIA) did not introduce noticeable differences. However, it was noted that geomagnetic activity during the studied period was very low and that MEE could still be important during more perturbed periods. The SEA we have performed was done on dates
with strong geomagnetic activity, representative of a doubling of normal activity. The AE index used here is however only a proxy for particle precipitation, and as such does not tell us for certain whether MEE were present during these days. A similar epoch analysis, performed on dates with only slightly enhanced geomagnetic activity, is used to provide a lower limit of the MEE contribution to the descending NO flux. Our analysis revealed that MEE can account for at least $4\%$ difference between descending NO levels.

Finally, one major aspect of the NO reservoir could play a key role in the NO winter descent: the altitude of the NO maximum density. This layer in WACCM is placed at a higher altitude throughout almost the entire year, with a six kilometre difference as compared to SOFIE during winter. Auroral electron precipitation in WACCM has a characteristic energy of 2 keV, corresponding to a maximum energy deposition at an altitude of 110 km, but increasing this characteristic energy does
not sufficiently lower the NO peak layer (Smith-Johnsen et al., 2018).

## 5    Conclusions

We investigated the ability of WACCM to simulate Antarctic NO concentrations in the MLT region and compared the results to SOFIE observations. The general features of the NO seasonal climatology are well captured by WACCM, though differences remain. Above the mesopause region, the modelled NO is almost a factor 2 higher in concentration and shows less seasonal
and inter-annual variability than observations. The NO maximum in WACCM is up to 6 km higher in altitude than in SOFIE. Using an MLR we have shown that a seasonal climatology and the NO variations from that climatology can explain more than $90\%$ of the variance in both datasets. The variations in NO are driven mainly by geomagnetic activity at high latitudes and the altitudinal profile of the geomagnetic driver is similar in WACCM and SOFIE. On the other hand, the impact of solar irradiance on NO, which is expected to be small at the polar regions, appears to be too large at high altitudes in WACCM and is linked to
the use of NOEM as upper boundary condition.

While the day-to-day geomagnetic activity thus drives NO variations in a similar way in WACCM and SOFIE, there are differences in the direct impact on absolute NO densities during strong geomagnetic disturbances. The maximum produced

NO was found to be consistently placed 5 km higher in WACCM than in SOFIE. During winter these NO enhancements descend with a remarkably consistent rate of about 2.2 km/day in the 80 - 110 km altitude region in both datasets, indicating that dynamical transport in the SH is accurately described in WACCM. The impact on the descending NO flux, however, is about twice as large in SOFIE and becomes progressively larger, up to a factor 4, lower in the MLT region, which indicates a

5   missing NO production process. We suggest three, possibly connected, mechanisms for the lower NO fluxes descending into the mesosphere: a too simplified parametrisation of D-region ion chemistry that can produce NO, excluded precipitation of medium energy electrons that directly produce NO and a too high altitude of the NO reservoir.

*Competing interests.* The authors declare no competing interests are present.

*Disclaimer.* TEXT

10  *Acknowledgements.* L. M. is supported by the Swedish Research Council under contract 621-2012-1648. D. R. M. is supported in part by NASA LWS grant NNX14AH54G. The National Center for Atmospheric Research (NCAR) is sponsored by the U.S. National Science Foundation (NSF). C. S.-J. was supported by the Norwegian Research Council under project 222390.

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

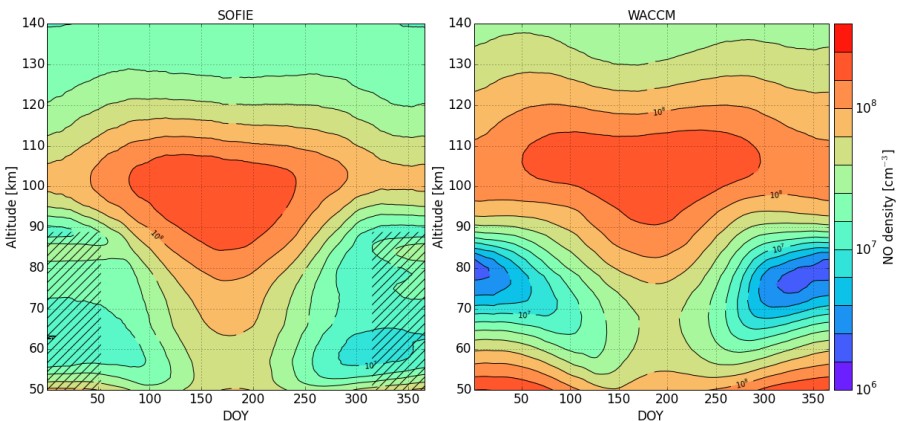

**Figure 1.** Seasonal climatology of Antarctic NO number density in SOFIE (left) and WACCM (right). Data are smoothed with a 3 month running average. Hashed areas occur during the Antarctic PMC season and should not be compared to the WACCM climatology (see more information in Section 2.1).

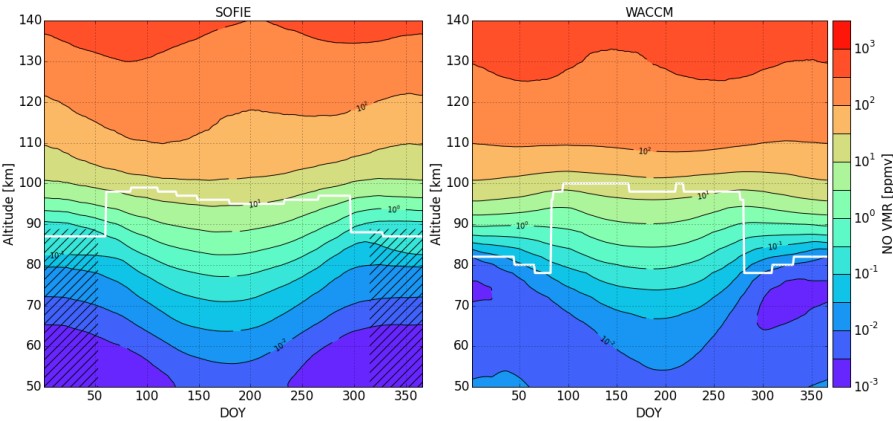

**Figure 2.** Seasonal climatology of Antarctic NO volume mixing ratio, similar as to Fig. 1. The white contour line represents the climatological mesopause altitude. Hashed areas occur during the Antarctic PMC season and should not be compared to the WACCM climatology (see more information in Section 2.1).

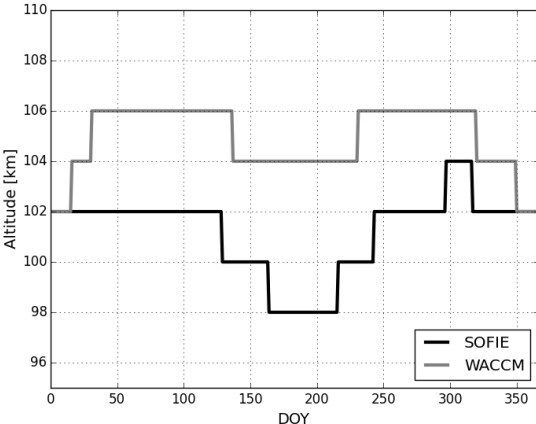

**Figure 3.** Altitude of the maximum NO number density obtained from the SOFIE and WACCM seasonal climatologies in Fig. 1.

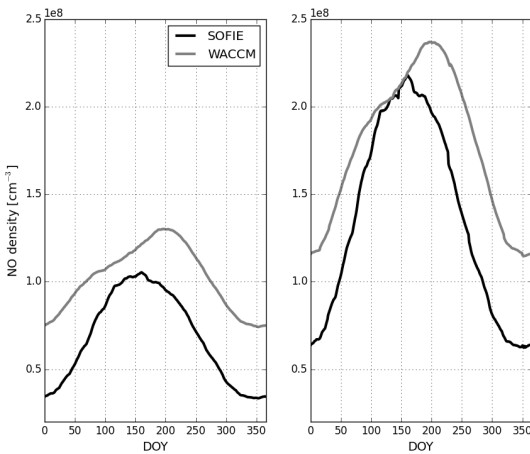

**Figure 4.** (left) Mean column density of NO in the lower thermosphere region from 90 to 140 km. (right) Mean column density in 10 km bin centred around the altitude of maximum NO.

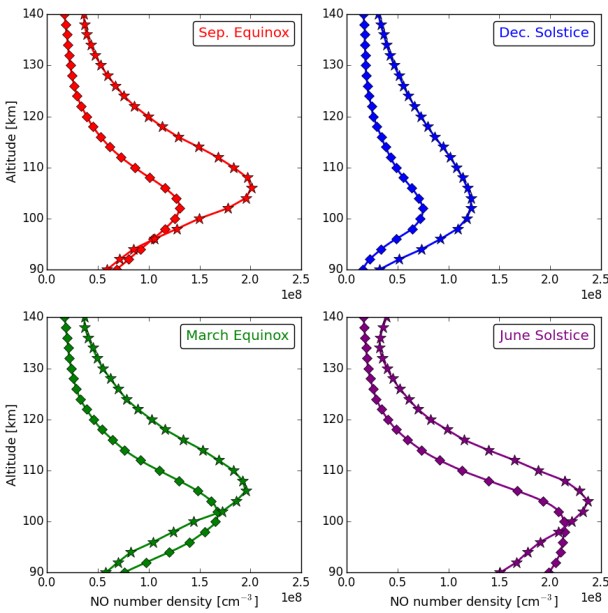

**Figure 5.** Seasonal variability of the lower thermospheric NO number density profile for SOFIE (diamonds) and WACCM (stars). Each season represents a multi-year mean of a 90 day period centred on the solstice or equinox. The September equinox and December solstice correspond to Antarctic spring and summer respectively, while the March equinox and June solstice correspond to the Antarctic autumn and winter season respectively.

**Table 1.** Selected dates during Antarctic winter on which the AE index increased more than 2 standard deviations.

| Year | Month-day |
|------|-----------|
| 2008 | 6-15, 7-13, 7-23, 8-10, 8-18 |
| 2009 | 5-07, 7-22, 8-30 |
| 2010 | 5-02, 5-29, 6-30, 8-04, 8-24 |
| 2011 | 5-28 |
| 2013 | 5-01, 7-14 |
| 2014 | 8-27 |

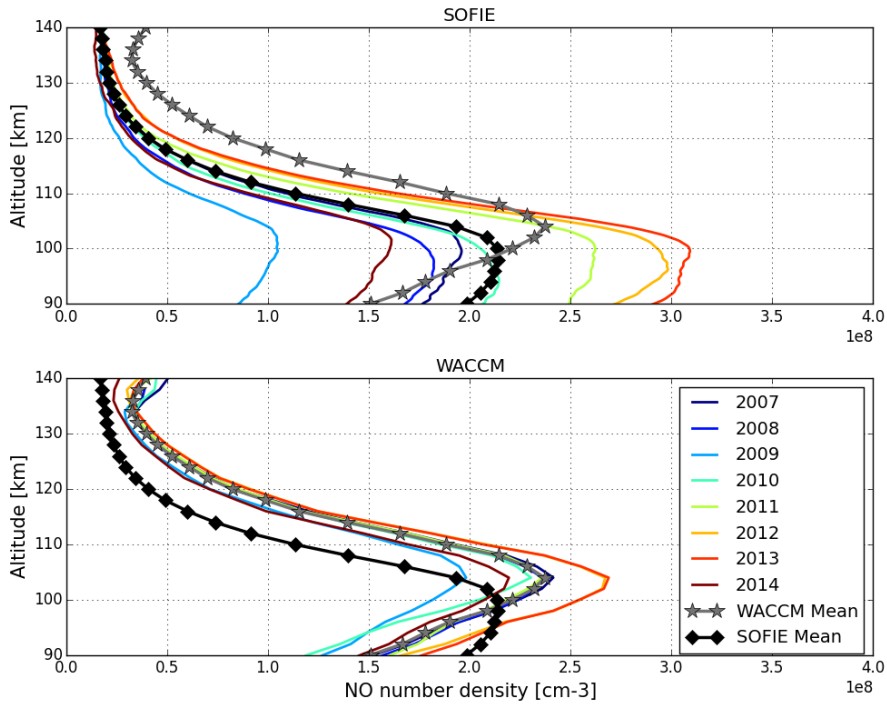

**Figure 6.** Inter-annual variability of the mean SH winter profile for SOFIE (upper) and WACCM (lower). A multi-year mean winter profile for SOFIE (black diamonds) and WACCM (grey stars) is given in each subfigure.

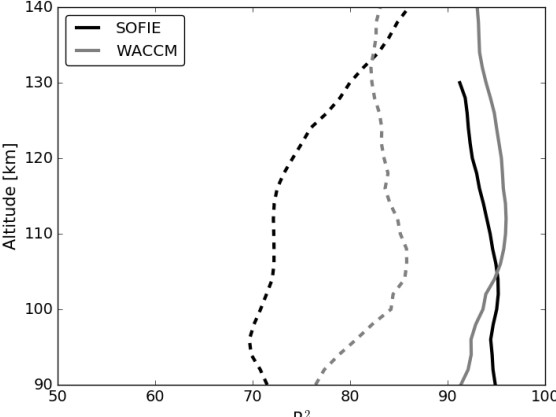

**Figure 7.** Percentage of the total variance in SOFIE (black) and WACCM (grey) data that can be explained by the seasonal climatology (dashed lines). Full lines represent the combined explained variance of the seasonal climatology and MLR model.

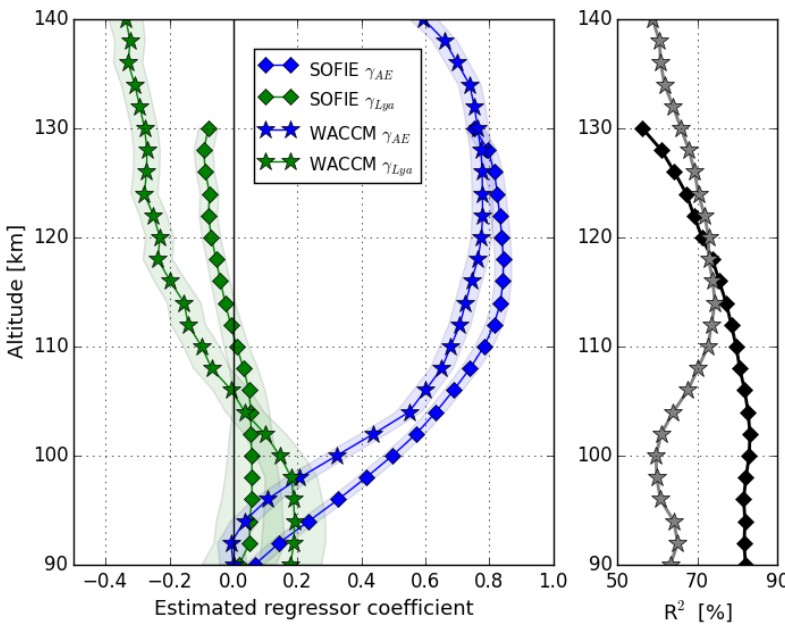

**Figure 8.** Results of the MLR performed on SOFIE (diamonds) and WACCM (stars) data throughout the lower thermosphere. (left) Estimates for the coefficients of geomagnetic activity (blue) and solar radiation (green), which can directly be compared to each other in terms of magnitude. (right) Total variation explained by the model for SOFIE (black) and WACCM (grey).

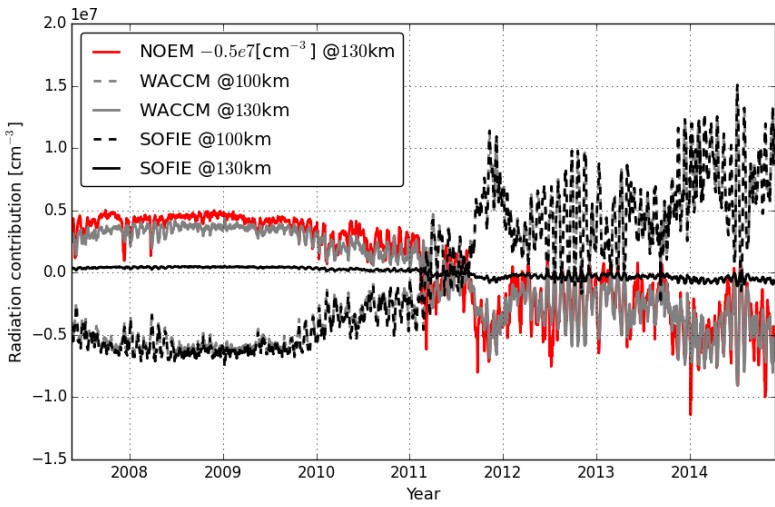

**Figure 9.** The contribution of Lyα radiation, as given by Eq. (4), to the NO budget for SOFIE (black) and WACCM (grey) at 100 km (dashed) and 130 km (solid) altitude. The solar contribution of NOEM at 130 km is shown in red for comparison to WACCM, is offset by a factor of $5.10^6$ cm$^{-3}$ and is based on the solar F10.7 radio index.

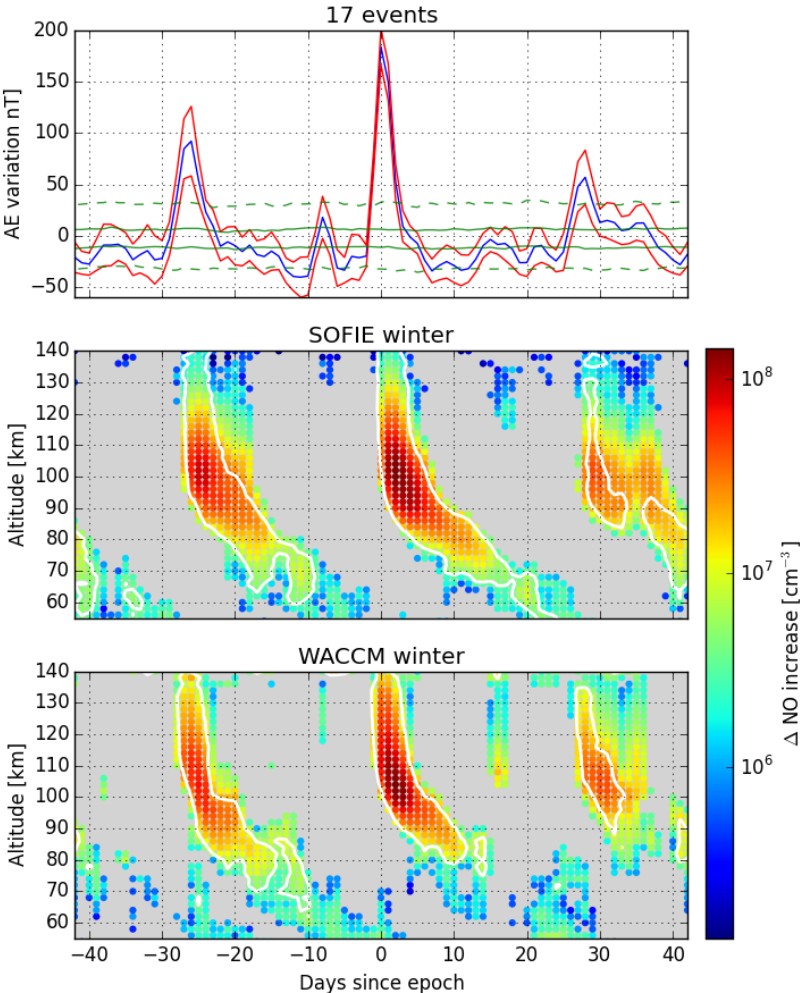

**Figure 10.** Epoch analysis performed every 2 km on winter hemispheric data in SOFIE (middle) and WACCM (lower). Dates are selected when the AE variation exceeds $2\sigma$ resulting in 17 events. (upper) Blue and red lines represent the mean and standard errors of the AE variations while full and dashed green lines represent $1\sigma$ and $2\sigma$ significance levels. (middle & lower) NO number density enhancements with the white contour line and the grey background representing a $1\sigma$ significance level and non-significant or negative NO variations respectively.

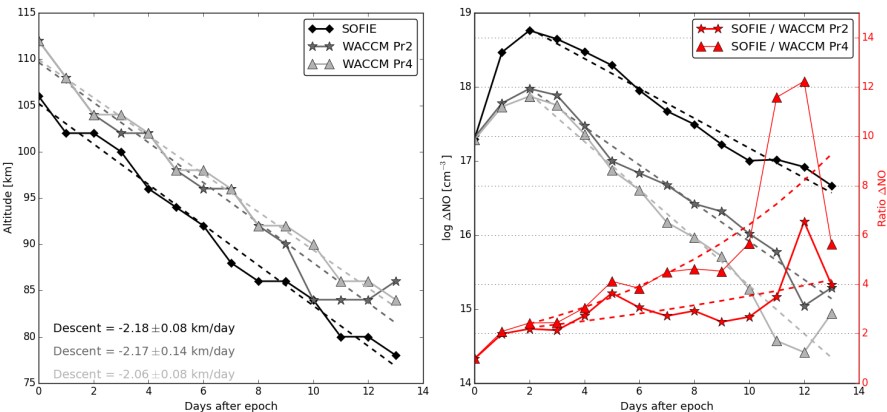

**Figure 11.** (left) Altitude of the maximum NO enhancement after the onset of geomagnetic activity for SOFIE and WACCM with enhanced diffusion (WACCM Pr2, obtained from Fig. 10) and for a control run with standard eddy diffusion (WACCM Pr4). The slope of a linear regression fit (dashed lines) represents the MLT descent rate. (right) The maximum NO enhancement at each corresponding day after the epoch, and at the corresponding altitude as shown in the left panel, that is transported downward (also obtained from Fig. 10). An exponentially decreasing fit (dashed black and grey lines) is performed onward from day 2, when the largest NO enhancement is reached. The ratio between SOFIE and WACCM NO enhancements (fit) is shown by the full (dashed) green line.

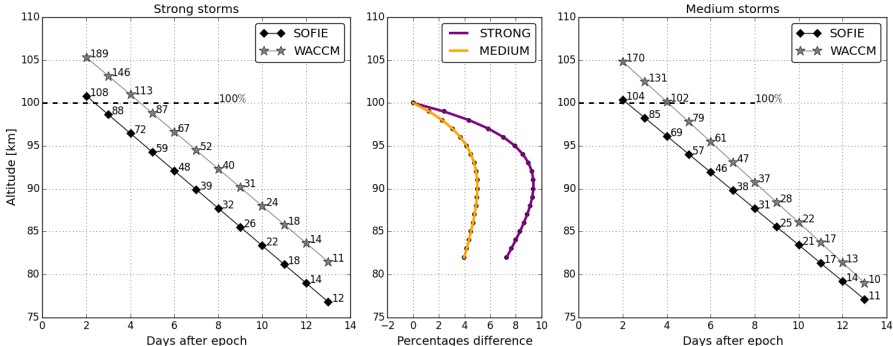

**Figure 12.** Percentage of NO for each day after the epoch that remains as calculated from the NO concentration at 100 km altitude for days with (left) strong and (right) medium geomagnetic activity. (middle) Difference of NO percentages between SOFIE and WACCM for each altitude.