# Peer review of "Production and transport mechanisms of NO in the polar upper mesosphere and lower thermosphere in observations and models"

_Atmospheric Chemistry and Physics, 2017_

## Referee Comment (RC1) · Anonymous Referee #2 · 26 Feb 2018

General:

The authors presents a detailed comparison of NO in the Antarctic MLT region between observations and simulations. They use SOFIE and WACCM data to study the magnitude of the NO reservoir, dependence on solar and geomagnetic drivers, and the descent through the mesopause. Similarities and differences and the reasons behind them are noted and discussed, and suggestions are given for further improvement and work.

Overall, the paper reads very well, the figures are clear except for some difficulty in separating black and dark green lines. The methods are sound and well described, the results and conclusions are supported by the data. The topic is clearly in the scope of ACP. Recommendation: publish after my concerns below are addressed.

[Figure]

One concern: altitude resolution and unit conversions. When describing the model and simulations, the authors mention that they interpolate the data to 2-km altitude grid. I think the WACCM grid is coarser than that in the MLT, so should not the observations be interpolated to the WACCM grid? Also, WACCM operates in pressure levels rather than altitudes. How was this conversion made? Also WACCM provides mixing ratios, but some results are shown as NO concentrations. How was this conversion made? The authors should provide some more details.

Another concern: differences in polar vortex dynamics. Since the polar NO is very much dependent on the polar vortex, I wonder what kind of differences are there between the reality and its representation in WACCM. SOFIE observations, as solar occultations, are very restricted in latitude. Thus sampling WACCM at the measurement locations could introduce artefacts if there is a SOFIE-WACCM difference in the shape or size of the vortex. Have the authors considered this possibility? I think that the problem, if any, could be largest during solstice times when lower latitudes are covered.

Specific:

The title is very general. Add: in the polar mesosphere-lower thermosphere. Maybe add SOFIE and WACCM. Maybe the years too.

Page 1, line 11. Maybe: altitude of peak density

Page 1, line 12. multiple linear regression

Page 5, line 23-27. Why are the observations giving different altitudes of maximum NO in the past and now? Is it due to instruments improving (e.g. better resolution) or the maximum altitude really changing? If the latter, then why the change?

Page 6, equation (1). Is AE the correct geomagnetic index to use? Why? In WACCM, auroral precipitation is driven by the Kp index, shouldn't that be used for the model at least? Is it possible that differences between Kp and AE could introduce an artefact?

Page 9, line 4-5. The relative increases given in the text are not presented in Figure 10, instead absolute values are shown. To me, the maximum absolute increases seem rather similar, so the difference in relative change is due to differences in the background?

Page 10, line 25. The N(2D)/N(4S) ratio is important, but I think it is in perhaps emphasised too much in general. There are other important factors, such as temperature and atomic oxygen. Model deficiencies in these could play a big role.

Page 12, line 29. "not parameterised chemistry in the D-region". Suggestion: excluded D-region ion chemistry. Or: too simplified parameterisation of D-region ion chemistry.
* * *

---

## Referee Comment (RC2) · Anonymous Referee #1 · 5 Mar 2018

The paper analyses NO production by energetic electron precipitation in the uppermost mesosphere and lower thermosphere from SOFIE observations, and in the WACCM model. This is an interesting investigation highlighting the importance of middle-energy electrons and possibly D-region ion chemistry for the upper mesospheric/lower thermospheric NO budget, and also separating the impact of UV/EUV radiation and electron precipitation. The study is interesting, though formulations sometimes could be a bit more precise, and the paper would profit from discussing results of previous studies by other authors more.

Title: it would be good to specify the altitude region here, i.e., "Production and transport mechanism of NO in the upper mesosphere and lower thermosphere in observations and models". I only really realized that the focus of the whole paper is the region above

75 km in the discussion; this should be clarified as early as possible, i.e., in the title.

Page 1, line 10: "the long term mean is too high": this is only true in the thermosphere, at altitudes above 110 km - see Fig.1. In the upper mesosphere and lower thermosphere – in 60-100 km – WACCM NO is quite considerably too low. Please be specific here, and discuss both the thermosphere and mesosphere.

Page 1, line 16-17: .... which is likely due to "missing" medium energy electrons and D-region "ion" chemistry.

Page 2, lines 4-5: there have been a large number of studies, both from observations and models, that investigate the impact of energetic electron precipitation on stratospheric ozone, radiation budget, and surface temperatures. You really should reference more of them here - the impression is that the Seppaelae et al 2013 study is the only one, or summarizes other studies - both not true. Here is a list of what I would reference: Natarajan et al., 2004; Randall et al., 2005; Schmidt et al., 2006; Marsh et al., 2007; Randall et al., 2007; Lu et al., 2008; Seppaelae et al., 2009; Baumgaertner et al., 2009; Reddmann et al., 2010; Semeniuk et al., 2011; Rozanov et al., 2012; Fytterer et al., ACP, 2015; Damiani et al., 2017; Sinnhuber et al., 2018.

Page 2, line 14: also reference Funke et al., 2014 because the MIPAS dataset really shows best how EPP NOy is transported down into the stratosphere in every winter, in both hemispheres.

Page 2, line 19: a 27 day periodicity in NO is also observed in SCIAMACHY data (Sinnhuber et al., 2016). This is at slightly lower altitudes, but should be discussed here anyway.

Page 2, line 20: also reference Randall et al., 2006; 2009; Funke et al., 2014; Sinnhuber et al., 2014; Funke et al., 2017

Page 2, line 21: Jackman et al., 2001; Funke et al., 2011

Page 2, line 22-23: Sinnhuber et al., 2018 show a similar relation in the impact of SPEs

and the indirect effect on the stratospheric NOy budget (1-2 Gmol/hemisphere versus up to 4 Gmol/hemisphere)

Page 2, line 25 to Page 3, line 12: another comparison of modeled and observed NOy is given in Sinnhuber et al., 2018, comparing 10 years of MIPAS satellite observations with results from three global models, two high-top models, and one "medium-top" model driven with an upper boundary parameterisation. They find very good agreement between observations and the medium-top model even after sudden stratospheric warmings - the difference to the Funke et al study is (I think) that a special parameterization for NOy during elevated stratopause events was implemented. Anyway this study should be discussed here as well.

Page 4, line 8: what is the vertical resolution of SOFIE? Usually this is not the same as the retrieval altitude grid.

Page 5, line 1 and Page 5, line 5: there appears to be a confusion here as to whether the lower thermosphere, or the mesosphere and lower thermosphere, are investigated. As Fig. 1, which is discussed in Section 3.1, shows altitudes down to 50 km, this appears to be "mesosphere and lower thermosphere", but should be noted consistently here.

Page 5, lines 6 - 13: you should also discuss the comparison of mesospheric NO values in the climatology shown in Fig 1. I found it confusing that most of the discussion appears to be related to altitudes above 100 km, though this is not stated explicitly: below 100 km, WACCM underestimates NO number densities everywhere above 60 km, particularly during polar winter. This should be discussed here, but is not.

Page 5, line 10: "descent is can be seen" strike out the "is"

Page 5, line 11: during the presence of → in the presence of

Page 5, lines 12-13: "... an overall higher column density" ... but underestimation of NO in the lowermost thermosphere and mesosphere, in particular during polar winter.

[Figure]

Page 5, line 14: "middle and upper atmosphere" would be everything from the tropopause to at least the exobase, do you mean "middle and upper mesosphere"?

Page 5, line 25-26, the references for Sheese at al 2011 and Sheese et al 2013 are mixed up – the 2011 paper investigates OSIRIS.

Page 5, 30-31: . . . WACCM total density is higher around the equinoxes in March and September . . . but only above 110 km. Please be more precise here.

Page 5, last sentence, . . . the discrepancy of equinoctial NO . . . could be an indication that the model is too sensitive to changes in geomagnetic activity . . . but only above 110 km, not below, where this discrepancy is not observed (at least not from Fig.1).

Page 6, lines 17-21: in years with low geomagnetic activity, NO is overestimated by WACCM quite considerably. However, in years with high geomagnetic activity (2011-2013), NO is underestimated by WACCM. I'm not quite sure what this means – the background is too high, but the variability of the geomagnetic forcing too low? But it should be discussed here, and taken into account later (in the discussion of the MRA).

Page 6, line 33: multilinear regression analysis has been used for NO in the upper mesosphere and lower thermosphere before, e.g., by Marsh et al., 2005 (Snoe); Bender et al., 2015 (ACE-FTS, ODIN/SMR, MIPAS, SCIAMACHY). These should be mentioned here, and the regression coefficients should be compared to the results of Marsh et al and Bender et al.

Page 7, line 3: . . . whether the correct processes drive NO densities at high latitudes "in the model".

Page 7, line 7: "a larger portion of the NO density than SOFIE" → "a larger portion of the NO density than in the SOFIE climatology"

Page 7, line 17-18, discussion of Fig 8: the SOFIE AE MLR curve looks as if it was shifted downward compared to the corresponding WACCM curve. Seen the other way round, WACCM seems to miss something below 120 km.

Page 7, line 24: . . . suggesting that solar forcing "due to soft x-rays or UV photolysis" . . . geomagnetic forcing can be interpreted as part of the solar forcing, as the acceleration mechanism of the precipitating electrons involve solar wind streams.

Page 7, lines 16 and following, discussion of the MLR coefficients: again, the coefficients for solar irradiance and AE index should be compared to the study of Bender et al., 2015, who carried out a similar analysis using four satellite instruments.

Page 8, line 28: "the polar vortex causes transport of air from the lower thermosphere into the mesosphere . . ." I don't think this is entirely correct – the polar vortex does not extend into the thermosphere, so does not cause anything there (directly). Dissipating gravity waves drive both the turbulent mixing across the winter-time mesopause and the mesospheric branch of the global meridional circulation, which, in the lower branch of the winter hemisphere (stratosphere, possibly lower mesosphere), also includes the polar vortex. Please be more precise in your formulations.

Page 9, line 19-20: I would change the order in this sentence: A possible source of NO that is not included in the current model is ionization by medium energy electrons (MEE) and D-region ion chemistry (Andersson et al., 2016).

Page 9, line 22, to page 10, line 5: I like the method of following the maximum peak down as shown in Figures 11 and 12. It is certainly an interesting analysis. However, I do not think you can derive a percentage values of by how much WACCM underestimates NO due to missing MEEs, due to the following reason: production of NO by MEE will occur most likely on this or the next two days, but below the altitude where the maximum NO peak occurs, as can be seen in Figure 10. This MEE produced NO will also be transported down, with the same rate as the maximum peak – your analysis misses this portion of the NO budget. I think the number you derive – the 4% difference, or 4-10 % from the middle panel of fig 12 – emphasizes the difference due to different / missing photochemistry, possibly the D-region ion chemistry.

Page 10, line 14: "the Lya regressor more strongly impacts WACCM . . ." the regressor

doesn't impact anything (except possibly our perception). The UV/EUV radiation appears to have a stronger impact in WACCM than in the observations. Please be more precise in your formulations.

Page 10, line 27 and following, branching ratio of N from particle impact ionization: most of the estimates go back to Porter et al 1976. However, if I understood Porter et al correct, the ratio provided there is a "high electron energy" limit, and might not be applicable to the thermosphere, where electrons of lower electrons are absorbed.

Page 11, line 6-7: a new temperature dependent reaction rate . . . of which reaction?

Page 11, line 14-15: with the same descent rate in WACCM and SOFIE – in which altitude range?

Page 12, line 3-4: I do not think that the 4% are correctly attributed to MEE, see my comment above. Also: you should compare your results to the model study by Arsenovic et al., 2016, targeting the MEE impact.

Page 12, line 30: "a too high in altitude NO reservoir" seems not a correct expression to me. "a too high altitude of the NO reservoir" might work.

---

## Author Comment (AC1) · 8 Jun 2018

Reply to Anonymous Referee #2:

We thank the reviewer for taking the time to read the manuscript and provide critical and valuable feedback.

**Overall, the paper reads very well, the figures are clear except for some difficulty in separating black and dark green lines.**
We have changed the green colour to red for improved clarity.

**One concern: altitude resolution and unit conversions. When describing the model and simulations, the authors mention that they interpolate the data to 2-km altitude grid. I think the WACCM grid is coarser than that in the MLT, so should not the observations be interpolated to the WACCM grid? Also, WACCM operates in pressure levels rather than altitudes. How was this conversion made? Also WACCM provides mixing ratios, but some results are shown as NO concentrations. How was this conversion made? The authors should provide some more details.**
The WACCM grid becomes progressively coarser near the model top and pressure levels are the fixed, native grid WACCM operates on. A conversion to geometric altitude is made using the geopotential height parameter, output by the model. The resulting profile is not fixed in altitude as it depends on latitude and therefore needs to be interpolated onto a fixed grid. Because SOFIE has a fixed vertical resolution of approximately 2 km, we decided to use this grid.
Conversion to number density is done using the ideal gas law. More information and equations have been added the text.

**Another concern: differences in polar vortex dynamics. Since the polar NO is very much dependent on the polar vortex, I wonder what kind of differences are there between the reality and its representation in WACCM. SOFIE observations, as solar occultations, are very restricted in latitude. Thus sampling WACCM at the measurement locations could introduce artefacts if there is a SOFIE-WACCM difference in the shape or size of the vortex. Have the authors considered this possibility? I think that the problem, if any, could be largest during solstice times when lower latitudes are covered.**
We thank the reviewer for pointing to this possibility. WACCM outputs data on geolocations as close as possible to SOFIE measurements and model data should ideally be as similar to the observations. We have not considered polar vortex locations from SOFIE data. Other research into MLT descent using satellite observations that are not restricted to solar occultations also observe discrepancies with the simulated descending NOx flux. Furthermore, the inferred descent rate in our study in the 80-110 km region is remarkably similar in WACCM and SOFIE, leading us to believe that the dynamics are well represented in the model. This has been discussed in the new version of the manuscript.

**The title is very general. Add: in the polar mesosphere-lower thermosphere. Maybe add SOFIE and WACCM. Maybe the years too.**
We have changed the title into 'Production and transport mechanisms of NO in the polar upper mesosphere and lower thermosphere in observations and models' and added the covered years in the abstract.

**Page 1, line 11. Maybe: altitude of peak density**
We have changed this to 'altitude of peak NO density'.

**Page 1, line 12. multiple linear regression**
It is changed.

**Page 5, line 23-27. Why are the observations giving different altitudes of maximum NO in the past and now? Is it due to instruments improving (e.g. better resolution) or the maximum altitude really changing? If the latter, then why the change?**

We have checked the SOFIE data and do not find any strong change in the altitude of maximum NO throughout the 2007-2015 period. We are therefore inclined to believe this changing altitude is due to increased resolution of the observations, though cannot rule out a change due to physical variability.

**Page 6, equation (1). Is AE the correct geomagnetic index to use? Why? In WACCM, auroral precipitation is driven by the Kp index, shouldn't that be used for the model at least? Is it possible that differences between Kp and AE could introduce an artefact?**

The AE index is the physically more correct index to use as it represent particle precipitation over the polar regions and is better related with NO variability (Hendrickx et al., 2015). WACCM uses the Kp index and when performing the MLR with Kp on both WACCM and SOFIE, a difference, similar as when using the AE index, remains between the datasets. Because the AE index is physically more correct, we decided to use the AE index rather than the Kp index.

**Page 9, line 4-5. The relative increases given in the text are not presented in Figure 10, instead absolute values are shown. To me, the maximum absolute increases seem rather similar, so the difference in relative change is due to differences in the background?**

The colour bar in Figure 10 is on a logarithmic scale, so even though the increases seem rather similar, a small change in colour can be a large difference and that is why we additionally give the percentage increase. This relative increase is a combination of a high NO background and a too low variation with geomagnetic activity in WACCM.

**Page 10, line 25. The N(2D)/N(4S) ratio is important, but I think it is in perhaps emphasised too much in general. There are other important factors, such as temperature and atomic oxygen. Model deficiencies in these could play a big role.**

We have changed the text in this paragraph as to not emphasise this ratio so strongly.

**Page 12, line 29. "not parameterised chemistry in the D-region". Suggestion: excluded D-region ion chemistry. Or: too simplified parameterisation of D-region ion chemistry.**

We have implemented the suggestion.

---

## Author Comment (AC2) · 8 Jun 2018

Reply to Anonymous Referee #1:

We thank the reviewer for taking the time to read the manuscript and provide critical and valuable feedback.

**Title: it would be good to specify the altitude region here, i.e., "Production and transport mechanism of NO in the upper mesosphere and lower thermosphere in observations and models".**
We have changed the title into 'Production and transport mechanisms of NO in the polar upper mesosphere and lower thermosphere in observations and models'.

**Page 1, line 10: "the long term mean is too high": this is only true in the thermosphere, at altitudes above 110 km - see Fig.1. In the upper mesosphere and lower thermosphere – in 60-100 km – WACCM NO is quite considerably too low. Please be specific here, and discuss both the thermosphere and mesosphere.**
We have made clear this statement is about the lower thermosphere and added a discussion on the mesosphere in Section 3.1.

**Page 1, line 16-17: .... which is likely due to "missing" medium energy electrons and D-region "ion" chemistry.**
It is changed.

**Page 2, lines 4-5: there have been a large number of studies, both from observations and models, that investigate the impact of energetic electron precipitation on stratospheric ozone, radiation budget, and surface temperatures. You really should reference more of them here.**
We have added several of the suggested publications.

**Page 2, line 14: also reference Funke et al., 2014 because the MIPAS dataset really shows best how EPP NOy is transported down into the stratosphere in every winter, in both hemispheres.**
This paper is referenced in the new manuscript.

**Page 2, line 19: a 27 day periodicity in NO is also observed in SCIAMACHY data (Sinnhuber et al., 2016). This is at slightly lower altitudes, but should be discussed here anyway.**
This paper is discussed and referenced in the new manuscript.

**Page 2, line 20: also reference Randall et al., 2006; 2009; Funke et al., 2014; Sinnhuber et al., 2014; Funke et al., 2017**
The papers are referenced in the new manuscript.

**Page 2, line 21: Jackman et al., 2001; Funke et al., 2011**
The papers are referenced in the new manuscript.

**Page 2, line 22-23: Sinnhuber et al., 2018 show a similar relation in the impact of SPEs and the indirect effect on the stratospheric NOy budget (1-2 Gmol/hemisphere versus up to 4 Gmol/hemisphere)**
This paper is referenced in the new manuscript.

**Page 2, line 25 to Page 3, line 12:** another comparison of modeled and observed NOy is given in Sinnhuber et al., 2018, comparing 10 years of MIPAS satellite observations with results from three global models, two high-top models, and one "medium- top" model driven with an upper boundary parameterisation. They find very good agreement between observations and the medium-top model even after sudden stratospheric warmings - the difference to the Funke et al study is (I think) that a special parameterization for NOy during elevated stratopause events was implemented. Anyway this study should be discussed here as well.

The paper is now discussed and referenced in the new manuscript.

**Page 4, line 8:** what is the vertical resolution of SOFIE? Usually this is not the same as the retrieval altitude grid.

The vertical resolution is approximately 2 km and stated on page 4, line 1.

**Page 5, line 1 and Page 5, line 5:** there appears to be a confusion here as to whether the lower thermosphere, or the mesosphere and lower thermosphere, are investigated. As Fig. 1, which is discussed in Section 3.1, shows altitudes down to 50 km, this appears to be "mesosphere and lower thermosphere", but should be noted consistently here.

We clarified this and write now 'MLT NO'.

**Page 5, lines 6 - 13:** you should also discuss the comparison of mesospheric NO values in the climatology shown in Fig 1. I found it confusing that most of the discussion appears to be related to altitudes above 100 km, though this is not stated explicitly: below 100 km, WACCM underestimates NO number densities everywhere above 60 km, particularly during polar winter. This should be discussed here, but is not.

The NO climatology above and below the mesopause region is now discussed.

**Page 5, line 10:** "descent is can be seen" strike out the "is"

It is removed.

**Page 5, line 11:** during the presence of → in the presence of

It is changed.

**Page 5, lines 12-13:** ". . . an overall higher column density" . . . but underestimation of NO in the lowermost thermosphere and mesosphere, in particular during polar winter.

This is clarified.

**Page 5, line 14:** "middle and upper atmosphere" would be everything from the tropopause to at least the exobase, do you mean "middle and upper mesosphere"?

We have changed and clarified this statement into '… the considered altitude range'.

**Page 5, line 25-26,** the references for Sheese at al 2011 and Sheese et al 2013 are mixed up – the 2011 paper investigates OSIRIS.

The references are now correct.

**Page 5, 30-31: . . .** WACCM total density is higher around the equinoxes in March and September . . . but only above 110 km. Please be more precise here.

It is clarified.

**Page 5, last sentence, . . . the discrepancy of equinoctial NO . . . could be an indication that the model is too sensitive to changes in geomagnetic activity . . . but only above 110 km, not below, where this discrepancy is not observed (at least not from Fig.1).**
It is clarified.

**Page 6, lines 17-21: in years with low geomagnetic activity, NO is overestimated by WACCM quite considerably. However, in years with high geomagnetic activity (2011- 2013), NO is underestimated by WACCM. I'm not quite sure what this means – the background is too high, but the variability of the geomagnetic forcing too low? But it should be discussed here, and taken into account later (in the discussion of the MRA).**
It is now discussed in more detail and linked to the MLR results.

**Page 6, line 33: multilinear regression analysis has been used for NO in the upper mesosphere and lower thermosphere before, e.g., by Marsh et al., 2005 (Snoe); Bender et al., 2015 (ACE-FTS, ODIN/SMR, MIPAS, SCIAMACHY). These should be mentioned here, and the regression coefficients should be compared to the results of Marsh et al and Bender et al.**
The papers are now referenced.

**Page 7, line 3: . . . whether the correct processes drive NO densities at high latitudes "in the model".**
This is clarified.

**Page 7, line 7: "a larger portion of the NO density than SOFIE" → "a larger portion of the NO density than in the SOFIE climatology"**
This is clarified.

**Page 7, line 17-18, discussion of Fig 8: the SOFIE AE MLR curve looks as if it was shifted downward compared to the corresponding WACCM curve. Seen the other way round, WACCM seems to miss something below 120 km.**
It is discussed in the new manuscript.

**Page 7, line 24: . . . suggesting that solar forcing "due to soft x-rays or UV photolysis" . . . geomagnetic forcing can be interpreted as part of the solar forcing, as the acceleration mechanism of the precipitating electrons involve solar wind streams.**
This is clarified.

**Page 7, lines 16 and following, discussion of the MLR coefficients: again, the coefficients for solar irradiance and AE index should be compared to the study of Bender et al., 2015, who carried out a similar analysis using four satellite instruments.**
Bender et al. (2015) indeed perform an MLR, but use a different approach: they include seasonality and perform the analysis on absolute NO densities. In our approach, we use NO anomalies and scale all variables, which means a direct comparison of the estimated coefficients is not possible. We reference the paper earlier in Section 3.2, so the reader can find the study.

**Page 8, line 28: "the polar vortex causes transport of air from the lower thermosphere into the mesosphere . . ."** I don't think this is entirely correct – the polar vortex does not extend into the thermosphere, so does not cause anything there (directly). Dissipating gravity waves drive both the turbulent mixing across the winter-time mesopause and the mesospheric branch of the global meridional circulation, which, in the lower branch of the winter hemisphere (stratosphere, possibly lower mesosphere), also includes the polar vortex. Please be more precise in your formulations.

It is clarified.

**Page 9, line 19-20: I would change the order in this sentence: A possible source of NO that is not included in the current model is ionization by medium energy electrons (MEE) and D-region ion chemistry (Andersson et al., 2016).**

It is changed as suggested.

**Page 9, line 22, to page 10, line 5: I like the method of following the maximum peak down as shown in Figures 11 and 12. It is certainly an interesting analysis. However, I do not think you can derive a percentage values of by how much WACCM underestimates NO due to missing MEEs, due to the following reason: production of NO by MEE will occur most likely on this or the next two days, but below the altitude where the maximum NO peak occurs, as can be seen in Figure 10. This MEE produced NO will also be transported down, with the same rate as the maximum peak – your analysis misses this portion of the NO budget. I think the number you derive – the 4% difference, or 4-10 % from the middle panel of fig 12 – emphasizes the difference due to different / missing photochemistry, possibly the D-region ion chemistry.**

We thank the reviewer for this remark, but WACCM does not include MEE and performing the epoch analysis on 'strong' and 'medium' storms tries to filter out MEE occurrence in the SOFIE observations. In the 'medium' storms, the percentage difference between SOFIE and WACCM is therefore most likely related to the different / missing photochemistry of D-region ion chemistry, as the reviewer suggests. Because the percentage difference between SOFIE and WACCM increases when the epoch analysis is performed on 'strong' storms, we can relate this change most likely to the occurrence of MEE, providing a lower limit of the MEE contribution to the indirect effect on NO. Furthermore, altering the arbitrary altitude of 100 km up or down does not change the range of deficit percentages nor the level where it maximises. We have explained the procedure better in the new manuscript.

**Page 10, line 14: "the Lya regressor more strongly impacts WACCM . . ."** the regressor doesn't impact anything (except possibly our perception). The UV/EUV radiation appears to have a stronger impact in WACCM than in the observations. Please be more precise in your formulations.

It is clarified.

**Page 10, line 27 and following, branching ratio of N from particle impact ionization: most of the estimates go back to Porter et al 1976. However, if I understood Porter et al correct, the ratio provided there is a "high electron energy" limit, and might not be applicable to the thermosphere, where electrons of lower electrons are absorbed.**

We thank the reviewer for the suggestion. We rephrased this paragraph as to not emphasise this ration so strongly.

**Page 11, line 6-7: a new temperature dependent reaction rate . . . of which reaction?**

The $N(^2D) + O_2 \rightarrow NO + O$ reaction, which is now stated in the text.

**Page 11, line 14-15: with the same descent rate in WACCM and SOFIE – in which altitude range?**
In the 80 to 110 km altitude region, which is now stated in the text.

**Page 12, line 3-4: I do not think that the 4% are correctly attributed to MEE, see my comment above. Also: you should compare your results to the model study by Arsenovic et al., 2016, targeting the MEE impact.**
Please find our reply above. We referenced the Arsenovic study.

**Page 12, line 30: "a too high in altitude NO reservoir" seems not a correct expression to me. "a too high altitude of the NO reservoir" might work.**
It is changed as suggested.